# Pharmacogenomic Biomarkers and Their Applications in Psychiatry

**DOI:** 10.3390/genes11121445

**Published:** 2020-11-30

**Authors:** Heejin Kam, Hotcherl Jeong

**Affiliations:** College of Pharmacy, Ewha Womans University, Seoul 03760, Korea; ladykam@ewhain.net

**Keywords:** precision medicine, personalized medicine, pharmacogenomics, pharmacogenomic biomarkers, psychiatry, psychiatric disorders, epilepsy

## Abstract

Realizing the promise of precision medicine in psychiatry is a laudable and beneficial endeavor, since it should markedly reduce morbidity and mortality and, in effect, alleviate the economic and social burden of psychiatric disorders. This review aims to summarize important issues on pharmacogenomics in psychiatry that have laid the foundation towards personalized pharmacotherapy and, in a broader sense, precision medicine. We present major pharmacogenomic biomarkers and their applications in a variety of psychiatric disorders, such as depression, attention-deficit/hyperactivity disorder (ADHD), narcolepsy, schizophrenia, and bipolar disorder. In addition, we extend the scope into epilepsy, since antiepileptic drugs are widely used to treat psychiatric disorders, although epilepsy is conventionally considered to be a neurological disorder.

## 1. Introduction

The World Health Organization (WHO) estimates that about 25% of the population around the world will suffer from at least one mental disorder at some time in their lives [1,2]. Depression and anxiety are among the most common disorders, and these can affect people regardless of age, gender, ethnicity, or background. We do not fully understand what causes most cases of mental health impairment, but it is known that both genetic and environmental factors can often contribute to an individual’s predisposition to a particular disorder. In other cases, serious injuries or traumatic events cause psychological symptoms that persist for a long period of time [3].

Medications can be used in order to reduce the intensity of symptoms or treat several psychiatric disorders. A patient’s response to the many medications used to treat various psychiatric disorders can be highly variable [4]. Drug response is dependent on personal health risk factors (e.g., gender, age, liver and renal function, blood pressure, body fat, alcohol and drugs, and drug–drug interactions). In addition, genetic factors, i.e., individual’s unique genetic makeup, can affect drug response influencing both pharmacokinetic parameters by causing variable activity of the systems that are responsible for the absorption, distribution, metabolism, and excretion of the drug and pharmacodynamic parameters, like the mechanisms of action of the drug [5,6]. Pharmacogenomics (PGx) refers to the study of drug response as it relates to potential individual genetic variations.

For an increasing number of drugs, pharmacogenomic testing is available and used to pre-screen patients and help them in selecting drug choice and drug dose accordingly [4,7]. Now, more than 10% of medications that are approved by the U.S. Food and Drug Administration (FDA) provide pharmacogenomic information (PGx information) in their drug labeling. This proportion is gradually increasing as more pharmacogenomic biomarkers (PGx biomarkers) are discovered and validated.

There are solid reasons for pharmacogenomic testing (PGx testing). Some drugs are only effective for specific genotypes and the testing can avoid unpredictable, severe, and potentially fatal drug reactions. Furthermore, for some drugs, a patient’s ancestry is the essential consideration. For example, for carbamazepine, a commonly used antiepileptic drug, the FDA recommends that, if patients are descendants of genetically high-risk populations, they should take PGx testing for the presence of *HLA-B*15:02* before treatment [8,9,10]. Carriers of this variant, which is frequently found in Han Chinese descendants, are highly susceptible to the development of Stevens–Johnson syndrome and toxic epidermal necrolysis, which often lead to serious conditions, during the course of carbamazepine therapy. The *HLA-B* variant alleles are just one example of such adverse drug reactions (ADRs). In fact, there is a plethora of genetic variants that are associated with ADRs. As an evident example, carriers of a variant of MT-RNR1 (mitochondrially encoded 12S rRNA), an RNA-coding gene, are at high risk of irreversible hearing loss by a single dose of gentamicin [11,12].

For a growing number of drugs, PGx testing provide a means of optimizing the drug choice and drug dose. Drug labels include not only standard dosing information, but also guidelines for adjusting the drug dose or selecting an alternative drug, when necessary, based on a patient’s genetic makeup if gene-drug interrelationships are well understood. Dosing adjustment requirements or recommendations are mostly in variants of genes that encode drug-metabolizing enzymes or drug transporters [13]. Thus, PGx biomarkers in genetic variants that are important for interindividual variations in PK and PD have been very useful in the optimization of pharmacotherapy. Several independent institutions, including the FDA [14], the European Medicines Agency (EMA), the Clinical Pharmacogenetics Implementation Consortium (CPIC) [15], the Canadian Pharmacogenomics Network for Drug Safety (CPNDS) [16,17], and the Dutch Pharmacogenetics Working Group (DPWG), have provided instructions on how PGx testing results can be interpreted in terms of the drug choice and the drug dose [18,19,20].

Accumulated data are then noted to FDA and its Table of Pharmacogenomic Biomarkers in Drug Labeling is widely used as a standard guideline [14]. PGx information is only included on labels when it is useful to inform clinicians of the impact of genotype on phenotype—gene–drug interrelationships—or to indicate whether a PGx test is available for a particular medication. As of now, the Table of PGx Biomarkers includes 431 drug-biomarker pairs for 298 drugs across therapeutic areas. In addition, PharmGKB provides a comprehensive resource, in which evidence-based PGx knowledges are curated and disseminated by scientific team about how our body responds to medications [21]. Pharmacogenomic information is important: it can maximize drug efficacy and reduce/avoid drug toxicity. Currently, FDA’s Table of PGx Biomarkers describes PGx information for 35 psychiatric medications, as in Table 1. In addition, the Table of PGx Biomarkers includes PGx information for eight antiepileptic drugs (AEDs), as in Table 2.

## 2. CYP2D6 and CYP2C9 Genes

The cytochrome P450s (CYPs) comprise a large superfamily of a variety of enzymes that serve as major workhorses for metabolizing steroid hormones, lipids, toxins, and xenobiotics. The CYP superfamily genes encode enzymes that function as monooxygenases and catalyze the modification of about 25–30% commonly used drugs [22,23]. The *CYP* genes are quite polymorphic and they can lead to increased, decreased, or completely absent drug metabolism activity. Among these genes, *CYP2D6* is particularly important and heavily studied. More than 100 *CYP2D6* variants have been reported and catalogued at the Pharmacogene Variation Consortium database [24]. In addition to large numbers of single nucleotide polymorphisms (SNPs), other types of variations—gene deletions, duplications, copy-number variants, and pseudogenes that are close to the gene—make genotyping very challenging.

Many of these variants cause the enzyme to change activity at different levels. The level of CYP2D6 activity decides how an individual responds to the substrate drugs. A standard dosage of the drug may show inadequate efficacy in some individuals and serious toxicity in others. To name a few, the drug substrates of CYP2D6 include atomoxetine (a non-stimulant for ADHD), clozapine (an antipsychotic for schizophrenia), and venlafaxine (an antidepressant), among psychiatric medications, as in Table 1 [4,25]. For these drugs, standard doses will result in higher-than-optimal active levels when individuals have absent or deficient CYP2D6 activity. Thus, the risk of ADRs increases and it may result in treatment failure.

There are substantial variations in *CYP2D6* allele frequencies among different populations [26,27]. The wild-type CYP2D6*1 allele shows normal enzyme activity and the extensive or normal metabolizer phenotype. The CYP2D6*2, -*33, and -*35 alleles also belong to this group. Other alleles contain non-functional variant(s), which produce a non-functioning enzyme (*3, *4, *5, *6, *7, and *8) or a decreased-activity enzyme (*10, *17, *29, and *41) [28]. Intermediate and poor metabolizers are individuals who carry decreased and null CYP2D6 alleles, respectively. Notably, approximately 30% of Asians and Asian descendants are intermediate metabolizers. In these populations, the *10 allele with decreased activity is very common: about 40%, when compared with about 2% in Caucasians [29]. Thus, a large proportion of Asians belong to intermediate metabolizers than Caucasians [30]. The African and African American populations also show a large proportion of CYPD6 alleles having sub-optimal activity. The frequencies of the remaining alleles vary depending on the population [30,31,32]. In Caucasians, only small proportions (less than 10%) are poor metabolizers [30]. In contrast, approximately 40% are extensive/normal metabolizers who carry two copies of *1 allele [33,34,35]. CYP2D6 poor metabolizers show higher levels of amitriptyline (as an example of drug substrates) in the plasma, when compared with extensive metabolizers, after standard doses of amitriptyline are taken [36]. When individuals carry a CYP2D6 null variant, their risk of developing ADRs becomes, at least, moderately increased [37]. Because standard dosages may cause to ADRs in poor metabolizers, it is recommended to avoid many tricyclic antidepressants (TCAs) and, instead, take an alternative option, a drug that is not a substrate of CYP2D6 [38].

Interestingly, copy-number variants were also found in *CYP2D6* genes [24]. In other words, individuals who carry more than two copies of functional CYP2D6 alleles—three to 13 copies of *CYP2D6* active allele—have been reported. These carriers are CYP2D6 ultrarapid metabolizers. In the case of 13 functional copies, the rate was up to 17 times higher than for individuals with no active CYP2D6 enzyme [39]. If the drug substrate has increased rate of metabolism, then its active form will not be available and, thus, the therapeutic response will become poor.

## 3. Antidepressants—Depression

Depression (also known as major depressive disorder) is a common and serious psychiatric disorder that affects more than 350 million people each year across the world [1]. People with disabilities face high risk for depression and, inversely, this disorder is a major cause of disability. Antidepressants are medications for treating depression [40]. They regulate neurotransmitters, such as serotonin, norepinephrine, and dopamine, which are linked to mood and emotion. Antidepressants can also be used for other indications, such as pain, anxiety, and insomnia. Even if their FDA approval is not designated to treat attention-deficit/hyperactivity disorder (ADHD), antidepressants are often used in order to treat the ADHD of adults.

Depending on the drug target, the neurotransmitter and/or its metabolizing enzyme, antidepressants can be categorized into four major classes—two older ones, tricyclic antidepressants (TCAs) and monoamine oxidase inhibitors (MAOIs), and two newer ones, selective serotonin reuptake inhibitors (SSRIs) and serotonin norepinephrine reuptake inhibitors (SNRIs) Table 1 [41,42]. SSRIs include citalopram, escitalopram, fluoxetine, paroxetine, and sertraline. SNRIs include duloxetine and venlafaxine. Bupropion is another newer antidepressant that is widely used, which, as a norepinephrine-dopamine reuptake inhibitor, works in a different mechanism of action than either SSRIs or SNRIs. These newer antidepressants, SSRIs, SNRIs, and bupropion, make fewer ADRs than older antidepressants and they can be used to treat a broader class of depressive or anxiety disorders.

Individual response to some antidepressants varies for reasons not yet well understood. Once a patient starts to take antidepressants, it is advisable not to stop taking them without the recommendation of a doctor. Therefore, it is necessary to know which medicine works for them beforehand. Some antidepressants may result in more adverse effects than others. Common ADRs include diarrhea, nausea and vomiting, weight gain, sleepiness, and sexual problems.

*SLC6A4*, the serotonin transporter gene, is a well-characterized gene that is associated with the treatment outcome of depression [43,44,45,46]. The serotonin transporter has twelve transmembrane helices and one extracellular loop located between helices 3 and 4. This transporter functions to reuptake serotonin (5HT) into the presynaptic neurons. The 5’ promoter region, called 5HT-transporter-linked polymorphic region (5-HTTLPR), of the *SLC6A4* gene is polymorphic in its length [47]. 5-HTTLPR locates approximately 1,000 bp upstream from the start site of transcription and comprises of repetitions of 16 imperfect 22-bp repeat units. 5-HTTLPR produces either a short (*s*) allele or the long (*l*) allele, respectively, depending on the occurrence of a deletion or an insertion of 6–8 units. The *l* allele is 44 bp shorter than the *s* allele and it is associated with higher levels of transcription of the *SLC6A4* gene in human cell lines. Some studies suggest that the *l*-allele patients respond better to antidepressants, even if this needs to be further investigated [48,49].

### 3.1. Amitriptyline

Amitriptyline (brand name Elavil) is a TCA that is used to treat many psychiatric disorders, including obsessive-compulsive disorder [50]. TCAs mostly mediate their therapeutic effects through inhibiting reuptake of serotonin and norepinephrine and, thus, hold more neurotransmitters around the synaptic cleft. The main drawback of TCAs is that their side effects are common, since they can also inhibit other receptors, such as α1-adrenergic, histamine H1, and muscarinic acetylcholine receptors. Additionally, this is the reason why TCAs have been replaced by SSRIs, which are more specific. Nonetheless, TCAs remain very useful in treating specific classes of depression and other psychiatric disorders. Amitriptyline is mostly metabolized by the CYP2C19 and CYP2D6 pathways [37]. Whereas CYP2C19 converts amitriptyline to active metabolites, such as nortriptyline, which is also a TCA, CYP2D6-mediated metabolism results in the less active 10-hydroxy metabolite.

The FDA’s guideline on amitriptyline states that CYP2D6 poor metabolizers will have higher plasma levels of the medication than expected when normal doses are given. The FDA also recommends that monitoring TCA plasma levels is necessary whenever a TCA is co-administered with another medication, which is known to be a CYP2D6 inhibitor. The CPIC has also made recommendations that the dosages of TCAs need to be adjusted, depending on CYP2C19 and CYP2D6 genotypes [38]. CYP2D6 poor metabolizers need to avoid any TCA due to the potential risk of ADRs. For CYP2D6 intermediate metabolizers, it is recommended to reduce the dosage up to about 75% of the usual dose. CYP2D6 ultrarapid metabolizers are recommended for avoiding TCAs, because of the potential lack of therapeutic effect and to take an alternative medication that is not a CYP2D6 substrate. CYP2C19 poor metabolizers need to avoid tertiary amine drugs due to the potential risk of ADRs and to consider a drug substitute that is not metabolized via the CYP2C19 pathway [51]. For CYP2C19 ultrarapid metabolizers, it is recommended to avoid amitriptyline, as this tertiary amine drug is expected to make an insufficient therapeutic effect, and to take an alternative medication that is not a CYP2C19 substrate, such as nortriptyline or desipramine.

One important precaution in increasing amitriptyline dose for CYP2D6 metabolizers is the fact that, when the level of hydroxyl-metabolites increases, it can lead to cardiotoxicity [52,53]. Presently, the optimal range of hydroxy-metabolite levels in the plasma and how the overall effects of CYP2D6 and CYP2C19 genotypes can be expressed as an individual’s treatment outcome of amitriptyline are poorly understood [38].

### 3.2. Venlafaxine

Venlafaxine (brand name Effexor) is an antidepressant that is used to treat the depression, anxiety, and panic disorder [54,55]. Both venlafaxine and its major metabolite, desvenlafaxine, belong to the drug class of SNRIs. As the CYP2D6 enzyme catalyzes the transformation of venlafaxine into its active metabolite, O-desmethylvenlafaxine (ODV), individuals with reduced or absent CYP2D6 activity will have high levels of venlafaxine, but low levels of ODV in the plasma. This situation can occur if a concomitant drug is a CYP2D6 inhibitor or the *CYP2D6* gene contains decreased-function allele(s).

The FDA’s guideline on venlafaxine indicates that neither specific dose recommendations for CYP2D6 poor metabolizers nor dose adjustment in the case of coadministration of venlafaxine and any CYP2D6 inhibitor. Instead, the drug label includes the statement that imipramine partially inhibits the metabolism of venlafaxine, but the total concentration of active compounds (both venlafaxine and ODV) remains unchanged. On the other hand, the DPWG provides recommendations on venlafaxine dosing based on the CYP2D6 genotype. According to the guideline, CYP2D6 poor and intermediate metabolizers needs to use an alternative drug. When ADRs occur and an alternative medication is not available, then venlafaxine can be used after a dose reduction based on clinical progress. The DPWG recommends that CYP2D6 ultrarapid metabolizers need to take an increased dose of venlafaxine up to 50% from the standard dose or use an alternative drug when treatment effects of dose adjustment cannot be monitored, as they will have lower plasma levels in the sum of venlafaxine and ODV [56].

CYP2D6 genotype can affect individual’s risk of ADRs from taking venlafaxine. CYP2D6 poor metabolizers have higher levels of venlafaxine and lower levels of ODV, and this can be interpreted as a higher risk of side effects and a reduced efficacy of the medication, respectively [57]. Old people tend to be at high risk [58,59,60]. For poor metabolizers receiving venlafaxine, ADRs are reported to occur more frequently in the gastrointestinal system (e.g., diarrhea and vomiting) or in cardiovascular system (e.g., hypertension, tachycardia, and prolonged QTc interval) [61,62]. Preemptive CYP2D6 genotyping before venlafaxine administration can make a decision on personalized dosing, which, in combination with therapeutic drug monitoring, could shorten the time taken to find an optimal maintenance dose and prevent potential ADRs [60,61,63,64,65]. However, there are discrepancies in the benefits of routine CYP2D6 genotyping. Some studies suggest that changes of drug metabolism that are rooted from CYP2D6 genetic variants make insufficient effects on the therapeutic levels of venlafaxine and that CYP2D6 genotyping would not forecast the treatment efficacy of venlafaxine in patients with depression [66,67,68,69].

## 4. Anti-Anxiety Medications—Anxiety Disorder

Anti-anxiety medications help to alleviate the symptoms of anxiety, such as intense and persistent fear and worry and panic attacks [70]. Benzodiazepines are the most commonly used anti-anxiety medications for treating anxiety disorders [71]. In the case of social anxiety disorder (also known as social phobia), benzodiazepines are a common second-line treatment option (its first-line treatment option is SSRIs or other antidepressants). Short half-life benzodiazepines include β-blockers and lorazepam, and they are used for the short-term anxiety. β-Blockers make the effects on physical symptoms of anxiety. For the long-term treatment of chronic anxiety, buspirone is an option, rather than benzodiazepines. Like other medications, some anti-anxiety medications also result in serious ADRs. However, we do not understand what causes these side effects and, thus, none of these anti-anxiety medications are currently included in either FDA’s Table of PGx Biomarkers or PharmGKB drug label annotations.

## 5. Stimulants and Non-Stimulants—ADHD and Narcolepsy

Attention-deficit/hyperactivity disorder (ADHD) is a common psychiatric and neurodevelopmental disorder in children, which is estimated to affect 5% of children and 2.5% of adults around the world [72,73,74,75]. According to the Diagnostics and Statistical Manual of Mental Disorders, Fifth Edition (DSM-5) [76], which is widely used in the United States, ADHD is marked as age-inappropriate pattern of inattention and/or hyperactive-impulsive symptoms [77]. On the other hand, the International Classification of Diseases, Eleventh Edition (ICD-11), which is widely accepted outside the United States, refers to ADHD as a hyperkinetic disorder and defines it as impairment in all three domains of inattention, hyperactivity, and impulsivity. This definition is a more narrow scope than the DSM-5 definition [72,78]. Currently, there are five ADHD medications that are approved by FDA, which can be dichotomized into stimulants (methylphenidate and amphetamine) and non-stimulants (atomoxetine, clonidine, and guanfacine) [79,80]. In this section, amphetamine and atomoxetine are examined due to their significance in PGx, as in Table 1.

Narcolepsy is a psychiatric disorder of hypersomnolence, which is characterized by disturbed sleep, sleep paralysis, hallucinations, and direct onset of REM sleep from wakefulness [63,64,65]. Narcolepsy is categorized into type 1 and type 2. Type 1 narcolepsy is characterized by the presence of cataplexy and loss of orexin (hypocretin) neurons that activate histaminergic neuron, which control wakefulness [66], whereas type 2 narcolepsy is characterized by the absence of cataplexy and normal cerebrospinal fluid hypocretin levels. For narcolepsy treatment, modafinil, antidepressants, and psychostimulants, such as pitolisant and methylphenidate, are applied, depending on patient’s symptoms [66]. In this section, modafinil and pitolisant, which are relevant with PGx, are examined.

### 5.1. Amphetamine

Amphetamine is a central nervous system stimulant that is approved for the treatment of ADHD and it is also used to treat narcolepsy as a third-line therapy [81,82,83]. Amphetamine has two enantiomers (*d*-amphetamine and *l*-amphetamine) and it can be composed of various types of salts, including *d*-methamphetamine salts, dexamfetamine, and a *d*-amphetamine prodrug, lisdexamfetamine dimesylate. In the 1930s, it was first noted that amphetamine relaxes hyperactivity, despite its stimulant nature [84]. Later, Cochrane Database of Systematic Reviews investigated 19 studies and concluded that patients rated that all amphetamines alleviated the severity of ADHD symptoms, while physicians evaluated lisdexamfetamine and mixed amphetamine salts alleviated the severity [85]. Amphetamine increases synaptic levels of monoamines, such as dopamines and noradrenalines in the brain, by inhibiting reuptake and releasing them into the synaptic cleft [86]. Although metabolism of amphetamine has not been thoroughly studied, it is known that CYP2D6 catalyzes the ring hydroxylation in order to produce 4-hydroxy-amphetamine [82,87]. Therefore, it is possible that amphetamine metabolism may vary in populations [82].

### 5.2. Atomoxetine

Atomoxetine is a non-stimulant, which is approved to treat ADHD [88]. Although its mechanism is not clear, as atomoxetine is a SNRI, it is thought that atomoxetine increases the synaptic levels of monoamines, including norepinephrine and dopamine in the brain by inhibiting reuptake [86]. Atomoxetine is mainly metabolized by CYP2D6 and, thus, CYP2D6 poor metabolizers have higher exposure to atomoxetine than the extensive metabolizers due to higher plasma concentrations and slower elimination [89]. Although atomoxetine has a wide therapeutic window, CYP2D6 poor metabolizers or patients who are taking a CYP2D6 inhibitor, such as paroxetine, fluoxetine, and quinidine, may benefit from improved therapeutic outcomes or suffer from an increased risk of adverse effects [88,90]. Therefore, atomoxetine dosing should be adjusted and carefully monitored for CYP2D6 poor metabolizers or patients concomitantly administering a CYP2D6 inhibitor.

### 5.3. Modafinil

Modafinil is approved for improving wakefulness in adult patients with excessive sleepiness that is related to narcolepsy, obstructive sleep apnea (OSA), or shift work sleep disorder (SWSD) [91]. It is commonly used in narcolepsy to treat daytime sleepiness [66]. Modafinil inhibits dopamine transporters and norepinephrine transporters, and these catecholamine effects are thought to give the cognitive and behavioral effects, including arousal and activity-promoting effects [92]. According to in vitro studies, modafinil and its metabolite, modafinil sulfone, are CYP2C19 reversible inhibitors. Therefore, modafinil may increase CYP2C19 substrate drugs, such as phenytoin, diazepam, propranolol, omeprazole, and clomiphene [91]. CYP2C19 also provides an ancillary route for certain CYP2D6 substrates, such as TCAs (clomipramine and desipramine) and SSRIs. CYP2D6 poor metabolizers on medications, which are metabolized by CYP2C19 ancillary pathway, may increase the plasma level of CYP2D6 substrates [91].

### 5.4. Pitolisant

Pitolisant (brand name WAKIX^®^) is approved for the treatment of excessive daytime sleepiness (EDS) in adult patients with narcolepsy. As a psychostimulant, pitolisant is a histamine-3 (H3) receptor antagonist and inverse agonist, which activates histamine release [93,94]. Pitolisant is majorly metabolized by CYP2D6 and minorly by CYP3A4 to inactive 5-amino-valeric acid and then further metabolized by glucuronidation or conjugation with glycine [94,95]. The main metabolic enzyme CYP2D6 affects plasma concentration. CYP2D6 poor metabolizers are less able to metabolize pitolisant into an inactive metabolite, so they have higher plasma concentrations of active pitolisant. Therefore, dose reduction is recommended for CYP2D6 poor metabolizers [94]. According to the pitolisant FDA-approved drug label, for CYP2D6 poor metabolizers, pitolisant should be titrated to a maximum dose of 17.8 mg once daily after seven days of treatment, while the recommended dosage range for normal metabolizers is 17.8 to 35.6 mg daily [94].

## 6. Antipsychotics—Psychosis with Schizophrenia or Bipolar Disorder

Antipsychotic medications are used to treat psychosis with schizophrenia and bipolar disorder [96,97]. The word “psychosis” means a condition that affects how the brain processes. Patients might see and hear things that do not exist (hallucinations) and/or believe unrealistic things (delusions). Psychosis can be a physical symptom, such as substance abuse, or a mental disorder with extreme stress or trauma, such as schizophrenia, bipolar disorder, or very severe depression (also known as psychotic depression). Antipsychotic medications do not cure these conditions, but, instead, can alleviate these symptoms and improve quality of life.

Typical antipsychotic medications are also called neuroleptics or first-generation antipsychotics [98]. They were first developed in the 1950s and still remain useful in the treatment of many psychosis symptoms, especially when other medications do not function. However, typical antipsychotics have a high risk of ADRs. Atypical antipsychotic medications are also called as second-generation antipsychotics. Both typical and atypical antipsychotics function as dopamine antagonists, but atypical antipsychotics show weaker affinity for dopamine receptors and, instead, strong affinity for serotonin receptors, particularly 5-HT2A [98]. The first atypical antipsychotic drug, clozapine, was first identified in 1958, but its development and approval in the US took a long time and, in the 1990s, clozapine finally emerged as a successful medication of schizophrenia, regardless of its high cost. Common atypical antipsychotics show a broad range of action and include aripiprazole, olanzapine, paliperidone, risperidone, and ziprasidone, as in Table 1.

Dopaminergic receptors are predominant G protein-coupled receptors in the vertebrate central nervous systems and they are involved in a variety of neurological processes [99]. These processes affect motivation, cognitive behavior, memory and learning, pleasure, and coordination of voluntary movements. Thus, dopaminergic receptors are common neuropsychiatric drug targets and, in fact, many antipsychotics are dopaminergic receptor antagonists. Many atypical antipsychotics inhibit serotonin (5-HT) receptors in the brain, the 5-HT2A receptor in particular, which play a critical role in schizophrenia.

Inter-individual differences in response to antipsychotic medications vary in a wide range and it often takes several trials to find the best medication that fits. In addition, antipsychotics can cause many ADRs, including weight gain, low blood pressure, and uncontrollable movements. Typical antipsychotics can cause a condition, called tardive dyskinesia (TD), when taken in the long term [100]. TD causes stiff, jerky muscle movements of the face, mainly at the mouth, which a person cannot control. TD rarely occurs while taking atypical antipsychotics.

According to the WHO, schizophrenia is a chronic and severe neurodevelopmental disorder, and it affects 20 million people worldwide. Etiology of schizophrenia remains unclear, but genetic and environmental factors seem to work together [101]. The pathogenesis involves a variety of neurotransmitter systems, which include dopamine, noradrenaline, glutamate, GABA, and acetylcholine. Schizophrenia can lead to substantial morbidity and mortality. Although antipsychotics are a mainstay of schizophrenia treatment, its treatment outcomes are inadequate for many patients. Antipsychotics exert their therapeutic actions by blocking D2 dopamine receptors on the post-synaptic membrane in the brain. The heritability of schizophrenia is very high, as demonstrated by family, twin, and adoption studies. For example, twin studies show that, if one identical twin develops schizophrenia, then there is a 40–50% chance that the other twin also has it. However, the search for “schizophrenia genes” has been elusive [102,103].

One genome-wide association study in 2014 identified 108 SNPs that were associated with schizophrenia, 83 of which were new. Not surprisingly, most of these SNPs occurred in genes that are highly expressed in the brain. However, surprisingly, some associated SNPs were found in genes that are expressed in tissues that are involved in the immune system [104].

Dysbindin-1 regulates the transport and release of synaptic vesicles and is encoded by the *DTNBP1* gene. Its genetic variants can affect the availability of dopamine D2 receptors. Recently, one study revealed that some genetic variants in the *DTNBP1* gene are associated with individual cognitive response to antipsychotics [105,106]. PGx association studies focused on pharmacodynamic genes in psychiatry are in progress and these should allow for better prediction of drug efficacy and potential ADRs.

### 6.1. Aripiprazole

Aripiprazole is an atypical antipsychotic that is used for managing schizophrenia, bipolar disorder, major depressive disorder, and Tourette’s disorder. Two enzymes, CYP2D6 and CYP3A4, are the mainstay of its metabolism and elimination. The FDA’s Table of PGx Biomarkers for aripiprazole states that CYP2D6 poor metabolizers need to take half of the usual dose [26]. When CYP2D6 poor metabolizers are taking concomitant strong CYP3A4 inhibitors (e.g., clarithromycin and itraconazole), the dosage should be adjusted as only one-quarter of the usual dose.

### 6.2. Clozapine

Clozapine is among the most effective antipsychotics available in treating schizophrenia and, in treatment-resistant schizophrenia, this is the only option to be effective. Schizophrenia patients show high risk of recurrent suicidal behavior and clozapine can function to reduce this condition [107]. When compared with typical antipsychotics, clozapine is much less likely to cause extrapyramidal side effects, which are severe movement disorders, but clozapine is administered to only the most severely ill, since there are substantial risks that are associated with clozapine therapy. Clozapine-induced agranulocytosis is one example and treatment with clozapine requires monitoring white blood and absolute neutrophil counts through enrollment requirements in a computer-based registry [108] Clozapine is mainly metabolized in the liver by CYP1A2 [109]. CYP2D6 and CYP3A4 enzymes are also involved in clozapine metabolism. CYP2D6 poor metabolizers may cause higher plasma levels of clozapine from usual doses than others. The FDA’s Table of PGx Biomarkers for clozapine states that a dose needs to be reduced for patients who are CYP2D6 poor metabolizers.

### 6.3. Risperidone

Risperidone, which is an atypical antipsychotic, is the most commonly prescribed among antipsychotics in the US [110,111]. It is used in treating schizophrenia, bipolar disorder, and severe dementia. CYP2D6 enzyme metabolizes risperidone to an active metabolite 9-hydroxyrisperidone. This reaction is also catalyzed by CYP3A4 to a lesser extent. Thus, CYP2D6 poor metabolizers would show a decreased ability to metabolize risperidone and be at a higher risk of ADRs due to an increased plasma level of risperidone than normal metabolizers who carry two active *CYP2D6* alleles. In contrast, CYP2D6 ultrarapid metabolizers may show a decreased response to therapy. The FDA’s Table of PGx Biomarkers for risperidone states that poor and extensive/normal metabolizers show no significant differences in the rates of adverse effects. Additionally, the DPWG recently revised its dosing guidelines as “no action is needed” for CYP2D6 poor metabolizers taking risperidone.

A recent association study of *KCNH7* variants reveals that two SNPs in the *KCNH7/Kv11.3* gene were significantly associated with risperidone responses [112,113]. The *KCNH7/Kv11.3* gene encodes one subunit of the potassium voltage-gated channel Kv11 family.

### 6.4. Thioridazine

Thioridazine is particularly used for patients who have failed to respond to, or cannot tolerate, other antipsychotics [114]. Thioridazine tends to lengthen the QT interval in a dose-dependent manner. Drugs showing this potential are associated with life-threatening ventricular tachycardia, called “torsades de pointes”. The CYP2D6 enzyme mainly metabolizes thioridazine [115]. Given standard doses of thioridazine, individuals with reduced levels of CYP2D6 activity may cause higher drug levels in the plasma and increase the potential of cardiac arrhythmias. The FDA’s Table of PGx Biomarkers for thioridazine states that thioridazine is contraindicated in patients who have low levels of CYP2D6 activity and in co-administration with other drugs that inhibit CYP2D6, such as fluoxetine and paroxetine, or inhibit the metabolism of thioridazine.

## 7. Mood Stabilizers—Bipolar Disorder

Mood stabilizers are used mainly in treating bipolar disorder and mood swings and in augmenting the treatment effect of antidepressants, through decreasing abnormal activity in the brain [116,117]. Lithium is approved for treating mania and bipolar disorder and it provides anti-suicide benefits to individuals on long-term maintenance. Anticonvulsant medications were originally developed to treat seizures, but were found to also help control unstable moods. These can be used as mood stabilizers and valproic acid is an example of anticonvulsants commonly used as a mood stabilizer. Mood stabilizers can lead to several ADRs, including excessively high blood levels.

Epileptic seizures are a malfunction of the brain, like major depression, anxiety disorders, and other psychoses. Thus, epilepsy has psychiatric aspects, and some psychiatric disorders are highly associated with epilepsy. Patients with mood disorders tend to experience seizures of both kinds—epileptic seizures and non-epileptic seizures. This relationship seems to flow in both directions: mood disorders increase the risk of seizures and seizures increase the risk of mood disorders. This is why some mood stabilizers are also used as antiepileptic drugs, and these will be described in the next section.

## 8. Antiepileptic Drugs—Epilepsy

According to the International League Against Epilepsy (ILAE) and the International Bureau for Epilepsy [118], epilepsy is defined as “a disorder of the brain characterized by an enduring predisposition to generate epileptic seizures and by the neurobiological, cognitive, psychological, and social consequences of this condition. The definition of epilepsy requires the occurrence of at least one epileptic seizure”. Being characterized by recurrent seizures, epilepsy affected 65 million people worldwide [119]. Although epilepsy can be controlled by epileptic drugs, the response to antiepileptic treatment varies highly, depending on a patient due to genetic variation in genes causing or related with the epilepsy in pharmacokinetics and pharmacodynamics, as in Table 2 [120].

### 8.1. Brivaracetam

Brivaracetam is approved for partial-onset seizures in patients four years of age and older [121]. It binds to the synaptic vesicle glycoprotein 2A (SV2A) with high affinity. It regulates the release of neurotransmitters at the presynaptic terminal and reduces neuronal excitability [122]. The amide moiety is hydrolyzed in primary metabolism and then CYP2C19 further metabolizes in order to form the hydroxy metabolite [121]. CYP2C19 intermediate or poor metabolizers are less able to metabolize the drug, resulting in a high plasma concentration of Brivaracetam, which may increase the risk of adverse effects, including sedation, fatigue, dizziness, and nausea. Therefore, patients who are CYP2C19 poor metabolizers or patients using CYP2C19 inhibitors may require a dose reduction [121].

### 8.2. Carbamazepine

Carbamazepine is approved for partial seizures, tonic-clonic seizures, mixed seizures patterns, and pain control for trigeminal neuralgia [123]. It is a tricyclic compound, and it binds to voltage-dependent sodium channels, which blocks the repetitive firing of an action potential by reducing polysynaptic responses [88]. Blocking the sodium channels and L-type calcium channel inhibits the action potential generated in the epileptic seizures [88]. Although carbamazepine is widely used as the first-line treatment, adverse reactions, such as hypersensitivity, occur to 10% of patients [88]. Hypersensitivity reactions are largely cutaneous, ranging from mild reactions, such as maculopapular exanthema (MPE) to life-threatening reactions, such as Stevens–Johnson syndrome (SJS), toxic epidermal necrolysis (TEN), and drug reaction with eosinophilia and systemic symptoms (DRESS) [88]. The risk of hypersensitivity is known to be correlated by specific human leukocyte antigen (HLA) alleles, such as HLA-B and HLA-A [124]. Particularly, the HLA-B*15:02 allele, which is known to mediate the activation of cytotoxic T-lymphocytes, mainly contributes to causing carbamazepine-induced SJS/TEN [120]. When compared with the HLA-B*15:02 allele, the HLA-A*31:01 allele shows a weaker association with causing SJS/TEN, yet it is identified to induce all carbamazepine hypersensitivity phenotypes, including MPE, DRESS, and SJS/TEN [88]. While HLA-B*15:02 is mainly found in Southeast Asia, HLA-A*31:01 is widely distributed across the world [88]. Because of these side effects, the FDA requires HLA-B*15:02 for patients in which HLA-B* may be present like Asian-descent patients. As of HLA-A*31:01, patients that are known to carry HLA-A*31:01 should weigh the risks and benefits before starting the medication [123].

### 8.3. Clobazam

Clobazam is approved for adjunctive treatment of seizures associated with Lennox–Gastaut syndrome (LGS) in patients two years of age or older [125]. LGS is a complex, rare, and severe childhood-onset epilepsy with multiple and concurrent seizures, cognitive retardation, and slow spike-wave activity in electroencephalogram (EEG) [88]. Clobazam is a 1,5-benzodiazepine and GABA_A_ receptor agonist, which binds to postsynaptic GABA_A_ receptors in the brain, causes hyperpolarization, and creates an inhibitory signal [126]. It is primarily metabolized by CYP3A4 and, minorly, by CYP2C19 and CYP2B6 to an active metabolite, N-desmethyl clobazam (norclobazam). Norclobazam is further metabolized to 4′-hydroxy-N-desmethylclobazam mainly by CYP2C19 [125,127]. Thus, poor CYP2C19 metabolizers have a long-term effect as the effects of norclobazam continue. Moreover, it is known that clobazam clearance is decreased in poor metabolizers, which increases the level of norclobazam [127,128]. Thirty-eight different CYP2C19 alleles have been identified so far that can affect the enzyme’s activity [129,130]. Clobazam poor metabolizers are required in order to adjust dosage [125].

### 8.4. Diazepam

Diazepam is approved for various clinical uses, including anxiety disorders, symptomatic relief of acute agitation, tremor, hallucinosis, and skeletal muscle spasm. It is also indicated for use as an adjunctive therapy for convulsive disorder [131]. It is a benzodiazepine and GABA_A_ receptor agonist, which creates an inhibitory signal in the vertebrate central nervous system [131,132]. CYP3A4 and CYP2C19 primarily metabolizes diazepam to an active metabolite, N-desmethyldiazepam [131]. It is further hydroxylated by CYP3A4 to another active metabolite oxazepam [131]. Both active metabolites are mainly eliminated by glucuronidation [131]. Genetic polymorphisms on CYP2C19 and CYP3A4 can affect the clearance of the drug, creating inter-individual variation, according to the FDA-approved drug label for diazepam gel (Diastat) [131]. On the other hand, the label for diazepam oral formulations (Valium) does not indicate pharmacogenomic relationship and only states that there is a potentially relevant interaction between diazepam and compounds that inhibit certain hepatic enzymes (particularly, CYP3A and CYP2C19) [131].

### 8.5. Lacosamide

Lacosamide is a third-generation anticonvulsant, which is approved for partial-onset seizures in patients four years of age and older [133]. As a functionalized amino acid, lacosamide is believed to amplify the slow inactivation of voltage-gated sodium channels in order to stabilize hyperexcited neuronal membranes and suppress the repetitive neuronal firing [134]. CYP3A4, CYP2C9, and CYP2C19 are involved in metabolism to form a major inactive O-desmethyl metabolite [88]. Amongst them, the role of CYP2C19 is extensively studied that CYP2C19 poor metabolizers have lower plasma concentration of the inactive O-desmethyl metabolite and similar concentration of active lacosamide. Consequently, CYP2C19 poor and normal metabolizers showed similar clinical effects in lacosamide pharmacokinetics [133].

### 8.6. Oxcarbazepine

As a structural analogue of carbamazepine, oxcarbazepine shows a similar mechanism of action with carbamazepine by blocking voltage-gated sodium channels [135,136]. Oxcarbazepine is metabolized to its monohydroxy derivative (MHD), which is an active metabolite, while carbamazepine is rapidly reduced to an epoxide metabolite [137]. In addition, oxcarbazepine inhibits N/P- and R-type calcium channels, whereas carbamazepine inhibits L-type calcium channels [135]. In terms of PGx testing, similar to carbamazepine, potential patients of oxcarbazepine should get tested for HLA-B*15:02 due to the risk of SJS/TEN [99]. However, HLA-A testing is not required, as oxcarbazepine is known to have fewer adverse reactions than carbamazepine due to their distinct pharmacokinetic characteristics [138].

### 8.7. Phenytoin

Phenytoin is approved for the control of generalized-onset tonic-clonic (grand mal) and complex partial (psychomotor and temporal lobe) seizures and the prevention and treatment of seizures occurring during or following neurosurgery [139]. It reduces brain stem centers′ maximal activity, which are responsible for tonic phase by stabilizing excessive stimulation induced hyperexcitability and reducing post-tetanic potentiation at synapses [139]. Phenytoin has a narrow therapeutic range of serum concentration between 10–20 μg/mL [88]. If the serum concentration exceeds the therapeutic range, then the risks of side effects increase and so therapeutic drug monitoring (TDM) of phenytoin is necessary [88,140]. It is known that CYP2C9 and CYP2C19 affect the serum concentration of phenytoin, as both of the enzymes are involved in the metabolism of phenytoin [88]. CYP2C9 poor metabolizers have reduced clearance rates, which may increase the serum level of phenytoin, increasing the risk of side effects. Similar to carbamazepine and oxcarbazepine, HLA-B*15:02 allele carriers have high risk of phenytoin-induced SJS/TEN, and so they should avoid phenytoin as an alternative for carbamazepine [139].

### 8.8. Valproic Acid

Valproic acid is approved for monotherapy and adjunctive therapy of complex partial seizures, simple and complex absence seizures, and adjunctive therapy of multiple seizure types that include absence seizures [141]. It is a fatty acid and enhances GABA transmission by inhibiting reabsorption of GABA and inhibits voltage dependent sodium and T-type calcium channels [132,142]. Valproic acid is contraindicated for patients with polymerase-γ-producing mitochondrial disease, as the genetic mutations in mitochondrial DNA polymerase γ (*POLG*) gene is strongly associated with acute liver failure and consequent death [141,142]. Valproic acid overexpresses *POLG1* gene and it changes several mitochondria genes, which affect mitochondria genesis [142]. Therefore, POLG mutation screening should be performed prior to valproic acid treatment [141]. Additionally, valproic acid is contraindicated for patients with urea cycle disorders (UCD) due to an increased risk of hyperammonemic encephalopathy, which can be fatal. Therefore, patients should be evaluated for UCD if they have a history of unexplained encephalopathy or coma, or cyclic vomiting and lethargy, or a family history of UCD, or show symptoms of UCD [141].

## 9. Beyond Pharmacogenomic Testing

Precision medicine targets providing the optimal diagnoses and treatments for each patient based on the categorization of biomarkers [143,144,145,146]. PGx is one of the main research areas of precision medicine. Nowadays, advances in artificial intelligence (AI), machine learning, multi-omics, and neuroimaging allow for analyzing and integrating complex genomic and clinical data in psychiatry and neurology. Artificial intelligence is the field of computing science that produces an algorithm based on available data to create predictive outcomes, even for unknown data in the future [147,148,149,150]. Particularly, state-of-the-art technology of deep learning revolutionized bioinformatics and medical imaging by yielding helpful software tools [151,152]. Whereas cancer therapy has routine clinical settings with well-established genomic data, in neuropsychiatry, the relationship between PGx data and their clinical significances has not been fully studied. Thus, the usage of artificial intelligence remains limited in the field.

AI has been used in predicting diagnosis, treatment outcome, and prognosis. As of psychiatry and neurology, multiple studies have used models, including deep learning architecture, random forest, tree-based ensemble, elastic net, and linear regression in order to evaluate and predict lithium treatment response on major depressive disorder [153]. To predict prognosis of major depressive disorder, there are algorithms, such as Gaussian process algorithm, Deep Patient, DeepCare, and Doctor AI, which utilize electronic health records. For example, Deep Patient has forecasted psychiatric disorders, including ADHD or schizophrenia with high accuracy (AUC = 0.863 and AUC = 0.853, respectively) [154]. However, the technology is still at an infancy phase and there are many obstacles and limitations to overcome in order to apply it clinically. For example, each algorithm is developed and assigned for each disease and so it is difficult to apply it to other diseases. Moreover, the sample size of each algorithm is too small to apply to public [153].

Various research efforts are ongoing in order to improve diagnosis, prognosis, and treatment in neuropsychiatry: PGx data and their treatment outcomes have been collected to support data-driven clinical decision-making for the patient. To this end, relations between genetic variation and variable drug responses to psychiatric medications should be well established [153]. The use of AI and machine learning analyses to predict individual-specific responses to psychiatric medications is challenging, but well worth pursuing [153].

## Figures and Tables

**Table 1 genes-11-01445-t001:** Food and Drug Administration (FDA) pharmacogenomic biomarkers in drug labeling in psychiatry.

Drug	Type	Indication	Biomarker	FDA	FDA Labeling Sections	EMA
**Antidepressants**						
Amitriptyline	TCA	Depression	CYP2D6	Actionable	Precautions	
Amoxapine	TCA	Depression	CYP2D6	Actionable	Precautions	
Bupropion	NDRI	Depression	CYP2D6	Informative	Clinical Pharmacology	
Citalopram	SSRI	Depression	CYP2C19	Actionable	Dosage and Administration Warnings Clinical Pharmacology	
			CYP2D6	Informative	Clinical Pharmacology	
Clomipramine	TCA	Depression	CYP2D6	Actionable	Precautions	
Desipramine	TCA	Depression	CYP2D6	Actionable	Precautions	
Desvenlafaxine	SNRI	Depression	CYP2D6	Informative	Clinical Pharmacology	
Doxepin	TCA	Depression	CYP2C19	Actionable	Clinical Pharmacology	
			CYP2D6		Clinical Pharmacology	
Duloxetine	SNRI	Depression	CYP2D6	Actionable	Drug Interactions	Actionable
Escitalopram	SSRI	Depression	CYP2C19	Actionable	Adverse Reactions	
			CYP2D6	Informative	Drug Interactions	
Fluoxetine	SSRI	Depression	CYP2D6	Informative	Precautions Clinical Pharmacology	
Fluvoxamine	SSRI	Depression	CYP2D6	Actionable	Drug Interactions	
Imipramine	TCA	Depression	CYP2D6	Actionable	Precautions	
Nefazodone	SARI	Depression	CYP2D6	Informative	Precautions	
Nortriptyline	TCA	Depression	CYP2D6	Actionable	Precautions	
Paroxetine	SSRI	Depression	CYP2D6	Informative	Drug Interactions Clinical Pharmacology	
Protriptyline	TCA	Depression	CYP2D6	Actionable	Precautions	
Trimipramine	TCA	Depression	CYP2D6	Actionable	Precautions	
Venlafaxine	SNRI	Depression	CYP2D6	Actionable	Drug Interactions Use in Specific Populations Clinical Pharmacology	
Vortioxetine	SSRI	Depression	CYP2D6	Actionable	Dosage and Administration Clinical Pharmacology	Actionable
**Stimulants and non-stimulants**						
Amphetamine	Stimulant	ADHD	CYP2D6	Informative	Clinical Pharmacology	
Atomoxetine	Non-stimulant	ADHD	CYP2D6	Actionable	Dosage and Administration Warnings and Precautions Adverse Reactions Drug Interactions Use in Specific Populations Clinical Pharmacology	
Modafinil	WPA	Narcolepsy	CYP2D6	Actionable	Clinical Pharmacology	
Pitolisant	H_3_R antagonist	Narcolepsy	CYP2D6	Actionable	Dosage and Administration Use in Specific Populations Clinical Pharmacology	
**Antipsychotics**						
Aripiprazole	Atypical	Schizophrenia Bipolar disorder	CYP2D6	Actionable	Dosage and Administration Use in Specific Populations Clinical Pharmacology	Actionable
Aripiprazole lauroxil	Atypical	Schizophrenia	CYP2D6	Actionable	Dosage and Administration Use in Specific Populations Clinical Pharmacology	
Brexpiprazole	Atypical	Schizophrenia Very severe depression	CYP2D6	Actionable	Dosage and Administration Use in Specific Populations Clinical Pharmacology	Actionable
Cariprazine	Atypical	Schizophrenia Bipolar disorder	CYP2D6	Informative	Clinical Pharmacology	
Clozapine	Atypical	Schizophrenia	CYP2D6	Actionable	Dosage and Administration Use in Specific Populations Clinical Pharmacology	
Iloperidone	Atypical	Schizophrenia	CYP2D6	Actionable	Dosage and Administration Warnings and Precautions Drug Interactions Clinical Pharmacology	
Paliperidone	Atypical	Schizophrenia	CYP2D6	Informative	Clinical Pharmacology	
Perphenazine	Typical	Schizophrenia	CYP2D6	Actionable	Precautions Clinical Pharmacology	
Pimozide	Typical	Tourette syndrome	CYP2D6	Testing Required	Dosage and Administration Precautions	
Risperidone	Atypical	Schizophrenia Bipolar disorder	CYP2D6	Informative	Clinical Pharmacology	
Thioridazine	Typical	Schizophrenia Other psychotic disorders	CYP2D6	Actionable	Contraindications Warnings Precautions	

H_3_R antagonist = histamine H_3_ receptor antagonist; NDRI = norepinephrine-dopamine reuptake inhibitor; SARI = serotonin antagonist and reuptake inhibitor; SNRI = serotonin and norepinephrine reuptake inhibitor; SSRI = selective serotonin reuptake inhibitor; TCA = tricyclic and tetracyclic antidepressant; WPA = wakefulness promoting agent.

**Table 2 genes-11-01445-t002:** FDA pharmacogenomic biomarkers in drug labeling in epilepsy.

Drug	Type	Indication	Biomarker	FDA	FDA Labeling Sections	EMA
Brivaracetam	Inhibits synaptic vesicle SV2A protein	Epilepsy	CYP2C19	Actionable	Clinical Pharmacology	Actionable
Carbamazepine	Enhances sodium channel (rapid inactivation)Inhibits L-type calcium channel	EpilepsyBipolar disorder	HLA-B	Testing Required	Boxed WarningWarningsPrecautions	
HLA-A	Actionable	Warnings	
Clobazam	GABA_A_ receptor agonist	Epilepsy	CYP2C19	Actionable	Dosage and AdministrationUse in Specific PopulationsClinical Pharmacology	
Diazepam	GABA_A_ receptor agonist	Epilepsy	CYP2C19	Actionable	Clinical Pharmacology	
Lacosamide	Enhances sodium channel (slow inactivation)	Epilepsy	CYP2C19	Informative	Clinical Pharmacology	Informative
Oxcarbazepine	Enhances sodium channel (rapid inactivation)Inhibits N/P- and R-type calcium channel	EpilepsyBipolar disorder	HLA-B	Testing recommended	Warnings and Precautions	
Phenytoin	Enhances sodium channel (rapid inactivation)	Epilepsy	CYP2C9	Actionable	Clinical Pharmacology	
CYP2C19	Clinical Pharmacology
HLA-B	Warnings
Valproic Acid	Inhibits voltage-dependent sodium and T-type calcium channels Enhances GABA transmission	Epilepsy	POLG	Testing Required	Boxed WarningContraindicationsWarnings and Precautions	
Nonspecific (Urea cycle disorders)	Actionable	ContraindicationsWarnings and Precautions

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
