# Peer review of "Pharmacogenomic Biomarkers and Their Applications in Psychiatry"

_genes, 2020, doi:10.3390/genes11121445_

Round 1

Reviewer 1 Report

Summary:

Kam and Jeong provide a detailed and comprehensive review of how pharmacogenomics is currently being applied to psychiatric treatments. They introduce pharmacogenomics and how genetic variation can influence drug effectiveness and safety. Also introduced are groups that are integrating PGx into clinical practice. The authors review the major drug pharmacogene, CYP2D6, and discuss how genetic variation, or copy number variants, affect CYP gene function. The authors then describe several psychiatric conditions followed by discussing the most common drugs used to treat these conditions and the current PGx understanding for these drugs. The psychiatric conditions discussed include depression; anxiety disorder; ADHD; narcolepsy; psychosis; bipolar disorder and epilepsy. The authors conclude their manuscript with a discussion of how AI can contribute to increasing precision psychiatry.

Major comments:

This is a well-organized and well-written review. It should be understandable by novices and experts.

Minor comments:

1. Lines 50-52. A reference should be included for the MT-RNR1:gentamicin association.

2. Line 62. Please review the reference to the CPNDS.

3. Line 70. First sentence can be deleted as it is discussed at the end of the paragraph.

4. Line 81. Please include a generic description of the enzymatic activity catalyzed by cytochrome P450s.

5. Lines 133-134. There are several instances where drug names are hyperlinked. Is this deliberate?

6. Venlafaxine is converted to the active metabolite ODV by CYP2D6. The authors indicate that the DPWG suggests increasing venlafaxine dosing by 150% in CYP2D6 ultrarapid metabolizers. What is the rationale for this because it would suggest that higher concentrations of active ODV would be present in these patients? Is higher dosing to maintain steady state levels of the parent drug? This needs further elaboration in the text.

Reviewer 2 Report

The paper by Kam and Jeong gives an overview of the knowledge related to the pharmacogenomics in psychiatry. The paper is well written and I have only few minor comments or advices for ameliorating the paper.

General remarks:

I would suggest that a native English speaker or language professional reviews the manuscript prior to resubmitting the article.

Mainor remarks:

Line 41: I would suggest splitting the sentence in two: “There are solid reasons for pharmacogenomic testing (PGx testing). Some drugs are effective…”

Line 126: The authors state that antidepressants regulate mood by changing neurotransmitters. Is this really so? Please rephrase this statement.

Line 130-131: Please correct the sentence to state: “Depending on the drug target, the neurotransmitter and/or its metabolizing enzyme antidepressants can be categorized into four major classes…”

Line 141: Please correct “to not stop” into “not to stop”.

Line 145: I would suggest replacing “affecting” with “associated with”.

Line 154: Please add some references supporting your statement that “all the studies of unipolar or bipolar depressed Caucasian patients, at least nominally significant associations could be found between the long variant and antidepressant response to SSRIs.”

Line 279: Delete “primarily” from the sentence.

Line 285: To make it clearer I suggest adding “FOR CYP2D6 poor metabolizers PITOLISANT should BE titrated to a maximum dose of 17.8 mg once daily after 7 days OF TREATMENT…”

Line 301-303: Please rephrase the sentence to make it more understandable: “It can be a physical symptom such as drug abuse or a mental disorder such as schizophrenia, bipolar disorder, or very severe depression, also known as “psychotic depression”.

Line 313-320: The authors should clarify more these general mechanisms of action for typical and atypical antipsychotics and mention what is their similarity (binding to D2 receptors) and what are the differences.

Line 426: The authors state that “level of Brivaracetam exists high”. I suppose the authors wanted to state that a high concentration of Brivaracetam is maintained but this is not fully clear.

Line 441: Please delete “is” from the sentence: “…which is known to mediate the activation of cytotoxic T-lymphocytes, IS strongly associated with carbamazepine-induced SJS/TEN.”

Line 462: Please replace “in use” with “for use” in the sentence: “It is also indicated FOR USE as an…”

Line 499: Please delete “to” in the sentence: “It is known that CYP2C9 and CYP2C19 affect TO the serum concentration…”

Line 510: When you refer to DNA polymerase you should write “DNA polymerase-ϒ gene (POLG)”

Line 523: I would suggest rewriting the sentence so it states: “Artificial intelligence is the field of computing science which produces an algorithm based on AVAILABLE data to create predictive outcomes for unknown data IN the future [108-111].”

Line 526-529: Please rephrase the sentence to make it more understandable.

Line 542-546: Please rephrase the sentence to make it more understandable.

Line 546-548: The last statement is maybe a bit to optimistic. I would suggest softening it up a bit.

Reviewer 3 Report

This is a relevant review on a very important and timely topic. However, the most problematic issue is indeed with the references (not) cited and the literature discussed. Please see some specific comments here below:

- A lot of statements are missing references. For example, first paragraph has no reference at all. This must be fixed reporting proper literature when making a statement. Similarly, the third and fourth paragraphs of the introduction, the entire paragraph of “Anti-anxiety medications”, the paragraph of “antipsychotics” … and so on… must revised throughout.

- The review seems to be based mostly on already approved genetic markers, while more discussion should be included for the most updated findings in the literature that could be inserted in future legislation.

- The paragraph on algorithms and way to use different genetic markers in the real world should be expanded.

- The review want to address pharmacogenomics markers in psychiatric disorders. However, it mostly address literature only on pharmacokinetic markers while completely ignoring new and old pharmacodynamics markers that are recently appearing. For example, for antipsychotics please see Heide et al American journal of Psychiatry 2016; Scheggia et al., Nature Communications 2018; or Leggio et al Molecular Psychiatry 2019… and many many other. This must be addressed by the current review in order to give the most updated and relevant message.

Round 2

Reviewer 3 Report

The authors well addressed my previous concerns. They should only make a small correction in the new sentence they added, as it can be misleading (lines 351-353): “PGx association studies focused on pharmacodynamic genes in psychiatry have had only limited success and future vigorous studies are required to make these preliminary data applicable to clinical practice.”

There must be more pharmacodynamics studies I agree, but the previous studies done were not “un-rigorous” nor only “preliminary” as different cohorts and replication samples were used… so I would rephrase or maybe delete the second part of the sentence.

Author Response

Dear Reviewer,

I greatly appreciate your time and effort again. I revised the sentence in lines 351-353 as you pointed out. Now it reads:

"PGx association studies focused on pharmacodynamic genes in psychiatry are in progress and these should allow better prediction of drug efficacy and potential ADRs."

Thank you so much.

Sincerely,

Ho Jeong